# FusionProt: Fusing Sequence and Structural Information for Unified Protein Representation Learning

**Dan Kalifa**  *kalifadan@cs.technion.ac.il*
*Department of Computer Science*
*Technion Israel Institute of Technology*

**Uriel Singer**  *urielsinger@meta.com*
*Meta AI*

**Kira Radinsky**  *kirar@cs.technion.ac.il*
*Department of Computer Science*
*Technion Israel Institute of Technology*

**Reviewed on OpenReview:** *https://openreview.net/forum?id=imcinaOHod*

## Abstract

Accurate protein representations that integrate sequence and three-dimensional (3D) structure are critical to many biological and biomedical tasks. Most existing models either ignore structure or combine it with sequence through a single, static fusion step. Here we present FusionProt, a unified model that learns representations via iterative, bidirectional fusion between a protein language model and a structure encoder. A single learnable token serves as a carrier, alternating between sequence attention and spatial message passing across layers. FusionProt is evaluated on Enzyme Commission (EC), Gene Ontology (GO), and mutation stability prediction tasks. It improves $F_{max}$ by a median of $+1.3$ points (up to $+2.0$) across EC and GO benchmarks, and boosts AUROC by $+3.6$ points over the strongest baseline on mutation stability. Inference cost remains practical, with only $\sim 2$–$5\%$ runtime overhead. Beyond state-of-the-art performance, we further demonstrate FusionProt's practical relevance through representative biological case studies, suggesting that the model captures biologically relevant features.

## 1 Introduction

Proteins are essential for biological processes and for understanding complex mechanisms in living organisms. They comprise linear chains of amino acids that fold into a specific three-dimensional (3D) structure, which underscores their functional diversity and dynamic behaviors (Zhang et al., 2023b). An effective understanding of proteins is essential for understanding disease mechanisms and synthetic biology and for advancing drug development (Liu et al., 2024).

Current methodologies for protein representation primarily emphasize the exploration of proteins' one-dimensional (1D) structures, specifically the relationships between amino acids. These approaches, such as ProteinBERT (Brandes et al., 2021) and ESM (Rives et al., 2019; Lin et al., 2023), often utilize text-based techniques and transformer architectures (Vaswani et al., 2017), which are trained on extensive protein sequence datasets. These models take an amino acid sequence as input and typically produce protein representations by averaging the representations of individual amino acids. However, this narrow focus on amino acid sequences neglects crucial protein structure details, thereby limiting the effectiveness of these methods.

The intricate 3D structure of proteins is crucial, as the conformation plays a pivotal role in determining their activities (Zhang et al., 2023a). Proteins possess specific active sites for interactions with other molecules,

which are defined by the 3D arrangement of amino acids. This 3D structure determines the specificity and affinity of binding interactions, aspects that a 1D representation cannot adequately capture (Kastritis & Bonvin, 2013; Yan et al., 2013). Furthermore, drug design relies on an understanding of 3D structures to identify binding sites and find molecules that modulate protein function (Luo, 2022; Liu et al., 2022). One of the state-of-the-art (SOTA) techniques today for 3D protein representation is GearNet (Zhang et al., 2023b). This approach converts the 3D structure of a protein into a graph that captures its biological characteristics. Subsequently, graph neural network techniques (Kipf & Welling, 2017; Schlichtkrull et al., 2017) are applied to this graph, facilitating the creation of comprehensive protein representations.

Recent research emphasizes the importance of comprehensive protein representation that includes both 1D and 3D structures to capture the protein's functional and interactional properties accurately. ESM-GearNet (Zhang et al., 2023a) was one of the first approaches to integrate these modalities. Although the study explored various fusion strategies, empirical results showed that the most effective method is using a large protein language model (PLM) such as ESM (Lin et al., 2023) to generate representations, which were then used as context for a graph encoder like GearNet (Zhang et al., 2023b). Other approaches, such as SaProt (Su et al., 2024), leverage an AlphaFold-based model (van Kempen et al., 2022) to reduce the 3D structure to tokens and train them along with amino acid tokens using a traditional PLM. However, these approaches are limited as they reduce one modality into context for another model, which potentially leads to the loss of critical structure information.

In this study, we introduce FusionProt (see Figure 1), a novel approach designed to learn a unified representation of the 1D and 3D structures of proteins simultaneously. Despite the vast number of proteins in nature, the number of known 3D structures remains limited (Váradi et al., 2023). To address this, we leverage an AlphaFold model (Jumper et al., 2021) for accurate protein structure predictions. We introduce an innovative learnable fusion token that serves as an adaptive bridge, enabling an iterative exchange of information between a PLM and the protein's 3D structure graph. This token is integrated into the training process of both modalities, enabling seamless propagation of information and facilitating comprehensive representation through iterative learning cycles. In practice, this token is concatenated to the sequence, allowing attention mechanisms to query the unique fusion token alongside the amino acids. This process extracts and integrates valuable information, enhancing the learning of amino acid representations. Then, the fusion token is incorporated as an additional node in the graph representing the protein's 3D structure, connected to all nodes. A graph encoder (i.e., a structure model) processes this graph, generating a new representation for the fusion token, which is subsequently used in the PLM training over the amino acids. Through this iterative process, the model representations are combined to form a refined protein representation.

Unlike prior models such as ESM-GearNet (Zhang et al., 2023a), which perform static or one-shot fusion, such as simple concatenation after independent encoding, FusionProt introduces a dynamic mechanism where the fusion token continuously evolves through repeated interaction across layers. This design allows sequence and structure modalities to co-adapt throughout training, resulting in richer, functionally informed representations that better capture the complexity of protein behavior.

We perform an empirical evaluation over several protein tasks, spanning a broad spectrum of biological domains, comparing numerous methods of protein representations and achieving SOTA performance across various benchmarks, with statistical significance improvements. We present ablation tests to better study the performance of the algorithm.

The contributions of this study are threefold: (1) We introduce FusionProt, a novel approach that learns a unified representation for both the 1D and 3D structures of proteins simultaneously. Our main focus is on the fusion of 1D and 3D models in an effective manner. Our method enhances the accuracy of capturing functional and interactional properties of proteins, addressing limitations of previous methods that treated these structures separately; (2) We propose a novel fusion architecture that utilizes a specialized learnable fusion token, enabling an iterative exchange of information between a PLM and the protein's 3D structure graph. This token is integrated into the training process of both modalities, enabling seamless propagation of information and facilitating comprehensive representation through iterative learning cycles. The iterative process facilitates the exchange of contextually relevant structure and sequential features, improving the model's ability to capture both 1D and 3D protein characteristics; (3) We conduct an empirical evaluation

of our work over several protein tasks establishing SOTA results and presenting biological case studies that further demonstrate the model's strengths. Also, we contribute our code to the community [1].

## 2 Related Work

### 2.1 Sequence-based Representation Learning

Proteins are comprised of sequences of amino acids, that establish a natural analogy to tokens in natural language processing. Recently, the adoption of unsupervised deep learning techniques has become prevalent in modeling protein sequence data.

The advent of Transformers (Vaswani et al., 2017) has led to the development of numerous PLMs such as MSA Transformer (Rao et al., 2021), ProteinBERT (Brandes et al., 2021), ProteinLM (Xiao et al., 2021), ProtBERT-BFD (Elnaggar et al., 2021), ProtTrans (Elnaggar et al., 2022), ESM-1b (Rives et al., 2019), and ESM-2 (Lin et al., 2023) which is considered as the SOTA PLM. These models were trained on data from UniRef (Suzek et al., 2007) which contains hundreds of billions of protein sequences, via masked language modeling (Devlin et al., 2019).

A protein's biological function hinges on its 3D native structure (Dill & MacCallum, 2012). However, many PLMs do not explicitly encode protein 3D structures, which are pivotal in understanding protein functions.

In this work, we aim to overcome this limitation by enhancing a PLM through novel fusion algorithms that integrate protein 3D structure models into the protein embedding. This approach seeks to capture both sequential and structure attributes of proteins, thereby advancing the capabilities of existing PLMs in biological research and applications.

### 2.2 Structure-based Representation Learning

The rising success of AlphaFold (Jumper et al., 2021; Senior et al., 2020) in predicting the 3D structure of proteins has led to deeper insights into their functional roles. Moreover, the release of more than 200 million protein structures in AlphaFoldDB (Váradi et al., 2023; 2021) has significantly advanced the development of large-scale protein structure models (van der Weg et al., 2025).

Protein structures are commonly represented as graphs, where amino acids serve as the nodes. Therefore, utilizing protein structure models on these graphs is a common practice. Models such as GVP (Jing et al., 2021), CDConv (Fan et al., 2023) and GearNet (Zhang et al., 2023b) have shown promising results, with different learning techniques. GearNet, which incorporates a Multiview Contrastive pre-training algorithm, is considered SOTA (Zhang et al., 2023b) and has outperformed the IEConv model (Hermosilla et al., 2021), which proposed to apply a learnable kernel function on edge features. Furthermore, CDConv (Fan et al., 2023) has outperformed HoloProt (Somnath et al., 2022) and ProNet (Wang et al., 2022a) in a recent study (Liu et al., 2023a).

Foldseek (van Kempen et al., 2022) suggested a different approach, optimized for protein structural search, which utilizes a VQ-VAE (van den Oord et al., 2017) to encode protein structures into informative tokens. Then, SaProt (Su et al., 2024) integrates residue and structure tokens during training, derived from encoding 3D protein structures using Foldseek.

In this study, we enhance the structure-based representation learning approach by integrating sequential information, thus capturing both sequential and structure attributes of proteins.

### 2.3 Joint Representation Learning

The integration of protein sequence-based models with protein structure models has gained popularity in recent years (Quan et al., 2024; Ko et al., 2024; Wu et al., 2022; Li et al., 2024).

---

[1] https://github.com/kalifadan/FusionProt

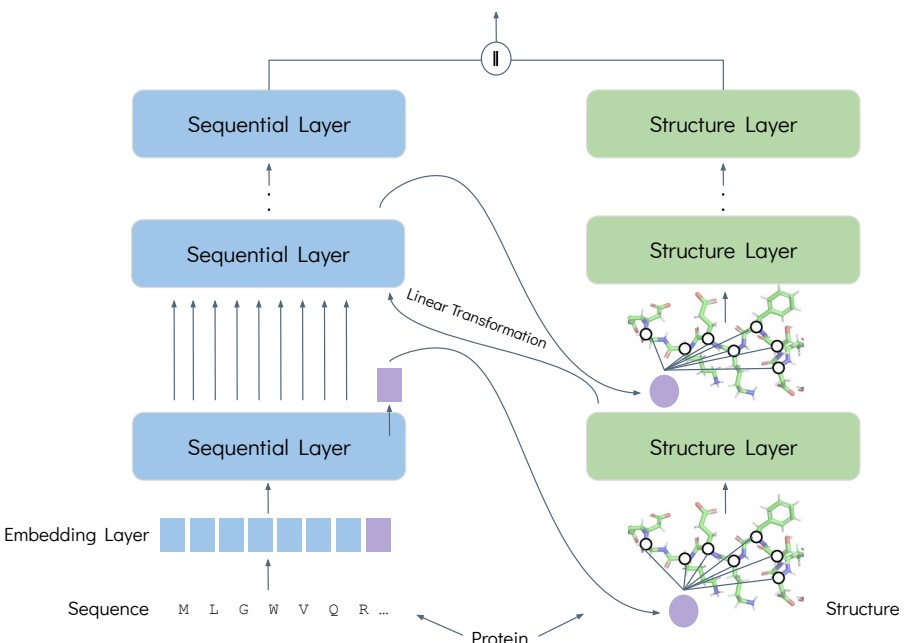

Figure 1: The **FusionProt** pre-training architecture. The model is trained on amino acid sequences and their corresponding 3D structures, utilizing the SOTA AlphaFold2 (Jumper et al., 2021) for accurate protein structure predictions. We introduce an innovative learnable fusion token that serves as an adaptive bridge, enabling an iterative exchange of information between a PLM and the protein's 3D structure graph. This token is concatenated to the protein sequence, allowing attention mechanisms to query the unique fusion token alongside the amino acids. Then, the fusion token is incorporated as an additional node in the graph representing the protein's 3D structure, connected to all nodes. A graph encoder (i.e., a structure model) processes this graph, generating a new representation for the fusion token. This representation is subsequently used in the PLM's sequential layers. A learnable linear transformation is applied between each sequential and structure layer pair to align and adapt their distinct modality spaces. Through this iterative process, the model representations are combined to form a refined protein representation.

Early efforts, such as LM-GVP (Wang et al., 2022b), SSEmb (Blaabjerg et al., 2024), MIF (Yang et al., 2022), and MIF-ST (Yang et al., 2022), aimed to combine PLMs with Graph Neural Networks (GNNs) (Scarselli et al., 2009). Related contrastive or alignment methods, such as RLA (Birnbaum et al., 2024) and S-PLM (Wang et al., 2025), learn to align sequence and structure embeddings but typically fuse modalities only at the representation level rather than within the token-level encoder.

More recently, ESM-GearNet (Zhang et al., 2023a) proposed to incorporate sequential information into distinct residue-level models such as GearNet, GVP, and CDConv (Zhang et al., 2023a). SaProt-GearNet (Su et al., 2024) also integrates GearNet and achieves similar results. However, these approaches are limited by reducing one modality into context for another model, potentially losing critical structure information.

Unlike earlier approaches, we introduce a novel fusion architecture that simultaneously learns a unified representation for both the 1D and 3D structures of proteins. To the best of our knowledge, no existing PLMs are based on an iterative fusion of structure and sequential models simultaneously.

## 3   Methods

We introduce the **FusionProt** model (see Figure 1). Given a protein $p = (S, R)$, which combines its amino acid sequence $S$ and its 3D structure $R$, FusionProt simultaneously learns a unified representation for both the 1D and 3D structures of the protein. We propose a specialized learnable fusion token designed to enable an iterative exchange of information between a PLM and the protein's 3D structure graph.

In this section, we formally define sequential (1D) and structure (3D) protein layers, which are used in the model as part of the protein representation learning, and present the FusionProt algorithm. Information from both layers is fused simultaneously during learning throughout the proposed specialized fusion token.

## 3.1 Protein Sequential Layer

### 3.1.1 Protein Sequence

The simplest level of protein structure, known as the primary structure, is a 1D structure, consisting of a linear sequence of amino acids, referred to as residues. The protein sequence exhibits similarities with natural language sequences, making the application of language models a common practice in this domain (Xiao et al., 2021; Brandes et al., 2021).

Given a protein sequence $S = [s_1, s_2, ..., s_n]$, where $n$ denotes the number of residues, the sequential model aims to capture the essential features of the protein sequence and outputs a protein representation denoted by $z = [z_1, z_2, ..., z_n] \in \mathbb{R}^{n \times D}$, where $D$ is the embedding dimension.

### 3.1.2 Sequential Layer Definition

We utilize ESM-2 (Lin et al., 2023) as our sequential model, leveraging its superior performance as a PLM. Alternatively, other advanced sequential models (i.e., PLMs) can be used.

Within the context of a given sequential layer denoted by $l$, we denote $z^{(l)}$ as the output representation of this layer, initialized with $z_i^{(0)} = \text{Embedding}(s_i) \in \mathbb{R}^D$, where $D$ denotes the embedding representation dimension. The layer performs the following update:

$$z^{(l)} = Attention\left(z_1^{(l-1)}, z_2^{(l-1)}, ..., z_n^{(l-1)}\right)$$

where $n$ denotes the number of residues in the protein, and *Attention* is a multi-head self-attention layer (Vaswani et al., 2017).

## 3.2 Protein Structure Layer

### 3.2.1 Protein 3D Structure

A protein 3D structure is uniquely determined by its primary structure (amino acid sequences) (Dill & Mac-Callum, 2012). There are only 20 standard residue (amino acid) types, each containing multiple components connected to a central carbon atom known as alpha carbon. Following GearNet (Zhang et al., 2023b), we use only alpha carbons to represent the main backbone structure of each protein. Therefore, we define a protein 3D structure as $R = [r_1, r_2...r_n] \in \mathbb{R}^{n \times 3}$, where $r_i$ represents the Cartesian coordinates of the $i-th$ alpha carbon atom of each amino acid, and $n$ denotes the number of residues, which is the sequence length.

### 3.2.2 Protein Structure Graph

We represent proteins using a multi-relational residue graph $G = (V, E, T)$, where $V$ is the set of residues, $E$ is the set of edges, and $T$ is the edge types. The set of edges $E$ consists of three directed edge types, namely sequential edges, radius edges, and KNN edges:

$$E_{\text{seq}} = \{(i, j) \mid i, j \in V, |j - i| < d_{\text{seq}}\}$$
$$E_{\text{radius}} = \{(i, j) \mid i, j \in V, |r_j - r_i| < d_{\text{radius}}\}$$
$$E_{\text{KNN}} = \{(i, j) \mid i, j \in V, j \in \text{KNN}(i)\}$$
$$E = E_{\text{seq}} \cup E_{\text{radius}} \cup E_{\text{KNN}}$$

where $d_{\text{seq}} = 3$ defines the sequential distance threshold, $d_{\text{radius}} = 10\text{Å} = 1$ [nm] defines the spatial distance threshold, and $\text{KNN}(i)$ indicates the K-nearest neighbors (Peterson, 2009) of node $i$ with $k = 10$ (all param-

eters were set as reported in GearNet (Zhang et al., 2023b)). In Section 5.5, we present an ablation study quantifying the impact of varying these values on predictive performance across the tasks.

### 3.2.3 Structure Layer Definition

We utilize GearNet (Zhang et al., 2023b) as the structure model, leveraging its contextual understanding of protein structures. Alternatively, other advanced structure models can be used in the algorithm (see Section 5.2).

Given a protein's 3D structure, $R$, we construct its corresponding protein structure graph $G = (V, E, T)$ (see Section 3.2.2). Then, the structure layer employs a relational message passing procedure, based on a relational graph convolutional neural network (Schlichtkrull et al., 2017).

Within the context of a given structure layer denoted by $l$, we denote $u^{(l)}$ as the output representations of this layer, initialized with $u_i^{(0)} = \text{Embedding}(v_i) \in \mathbb{R}^D$, where $v_i \in V$, and $D$ denotes the embedding representation dimension. The layer performs the following update:

$$u_i^{(l)} = u_i^{(l-1)} + \sigma \left( \sum_{t \in T} W_t \sum_{j \in \mathcal{N}_t(i)} u_j^{(l-1)} \right)$$

where $\mathcal{N}_t(i)$ is the set of neighbors of $i$ with edge type $t$, $\sigma(\cdot)$ is a ReLU activation function, and the weight matrix $W_t$ is learned per edge type $t$. This approach allows the model to incorporate various types of relational information, thereby enhancing its ability to learn comprehensive protein representations.

## 3.3 FusionProt

### 3.3.1 Fusion Design

We introduce a specialized learnable fusion token, designed to enable an iterative exchange of information between a PLM and the protein's 3D structure graph. This fusion token serves as a dynamic bridge, enabling iterative information exchange between the two modalities. During training (see Figure 1), the specialized fusion token holds information from both models, the sequential and the structure. In practice, the amino acids query the associated unique fusion token to extract and integrate valuable information for learning amino acid representations via attention mechanisms. Additionally, the fusion token is incorporated as an additional node in the graph representing the protein's 3D structure. We connect it to all nodes, with a new edge type, thus enabling the structure model to learn important features from the sequential model. Hence, the token enables a novel fusion between both models, leading to an enhanced protein representation.

Our choice to connect the fusion token to all residues follows GNN principles (Scarselli et al., 2009), enabling information exchange that captures both global and local dependencies. This universal reach provides single-hop access to any site, preventing bottlenecks in elongated or multi-domain proteins that arise with limited neighborhood expansion. The design is analogous to virtual nodes that aggregate and broadcast global context (Hwang et al., 2022; Xu et al., 2019; Zhang et al., 2025), but here the token mediates cross-modal communication between the PLM and the 3D graph rather than graph-only messaging. A complementary line, graph prompt learning, prepends learned prompt tokens to steer a single-modality GNN or a pre-trained graph model toward the task (Liu et al., 2023b; Zi et al., 2024). In contrast, our fusion token participates at every block in both PLM attention and GNN message passing, is instance-conditioned and updated layer-by-layer, and thus provides bidirectional routing while preserving modality-specific inductive biases (sequence order; 3D geometry).

The fusion token is not merely an addition to the sequence or structure; rather, it mediates the dynamic interaction between the two modalities, ensuring iterative refinement of the protein representation. This approach contrasts with previous methods that simply concatenate or independently process the sequence and structure information. This novel fusion mechanism leverages multi-modal learning and graph-based information propagation. By iteratively combining the 1D protein sequence and 3D structure features, FusionProt captures both local and global dependencies, enabling the model to learn richer and more holistic representations of proteins.

To provide transparency into this process, we visualize (see Supplementary Section A.3) fusion-token attention across layers and around each structure injection.

---

**Algorithm 1** FusionProt Algorithm

---

**Input:** Protein $p = (S, R)$ with sequence $S = [s_1, ..., s_n]$, and 3D structure $R = [r_1, ..., r_n] \in \mathbb{R}^{n \times 3}$, Sequential encoder with $L_1$ layers, Structure encoder $L_2$ layers.

**Output:** Unified protein representation $h$

1: Create a learnable fusion token $s_{n+1} = f$
2: Augment sequence: $S' = [s_1, ..., s_n, s_{n+1}]$
3: Initialize sequential embeddings: $z_i^{(0)} = \text{Embedding}(s_i) \; \forall i \in [1, ..., n+1]$
4: Construct graph $G = (V, E, T)$ from $R$
5: Add node $v_{n+1}$ to $G$, connect $v_{n+1} \leftrightarrow v_i$ with edge type $t_f \; \forall i \in [1, ..., n]$
6: Set $u_i^{(0)} = \text{Embedding}(v_i)$ for all $v_i \in V$
7: **for** $l = 1$ to $L_2$ **do**
8:     **for** $j = 1$ to $L_1 \div L_2$ **do**
9:         $l_j = j + (l - 1) \cdot (L_1 \div L_2)$
10:         $z^{(l_j)} = Attention(z_1^{(l_j - 1)}, ..., z_n^{(l_j - 1)}, z_{n+1}^{(l_j - 1)})$
11:     **end for**
12:     $u_{n+1}^{(l-1)} = \text{Linear}_l(z_{n+1}^{(l_j)})$
13:     Update node $v_{n+1}$ in graph $G$ with $u_{n+1}^{(l-1)}$
14:     $u_i^{(l)} = u_i^{(l-1)} + \sigma \left( \sum_{t \in T} W_t \sum_{j \in \mathcal{N}_t(i)} u_j^{(l-1)} \right) \; \forall u_i \in V$
15:     $z_{n+1}^{(l_j)} = \text{Linear}'_l(u_{n+1}^{(l)})$
16: **end for**
17: Concatenate final embeddings: $h = [z_{1:n}^{(L_1)}; u_{1:n}^{(L_2)}]$
18: **return** $h$

---

### 3.3.2 Fusion Algorithm

We provide a formal description of the FusionProt algorithm in Algorithm 1.

Given a protein $p = (S, R)$, which combines its amino acid sequence $S$ and its 3D structure $R$, FusionProt concatenates the learnable fusion token to the sequence, resulting in a new sequence $S' = [s_1, s_2, ..., s_n, f]$ with a length of $n + 1$. Then, it is used to initialize the sequential embedding layer:

$$\forall s_i \in S' : z_i^{(0)} = \text{Embedding}(s_i)$$

Within the context of a given sequential layer denoted by $l$, the algorithm yields a sequence representation:

$$z^{(l)} = Attention \left( z_1^{(l-1)}, z_2^{(l-1)}, ..., z_n^{(l-1)}, z_{n+1}^{(l-1)} \right)$$

where $z_{n+1}^{(l-1)}$ is the intermediate representation of the fusion token. Then, the fusion token representation is passed into the corresponding structure layer $l$, represented as node $n + 1$, while connecting to all nodes in the 3D structure graph, yielding a structure representation:

$$u_i^{(l)} = u_i^{(l-1)} + \sigma \left( \sum_{t \in T'} W_t \sum_{j \in \mathcal{N}_t(i)} u_j^{(l-1)} \right)$$

where $u_{n+1}^{(l-1)} = \text{Linear}_l \left( z_{n+1}^{(l-1)} \right)$, $T' = T \cup \{t_f\}$, $\mathcal{N}_{t_f}(n+1) = V$, and $\forall i \in [1, ..., n] : \mathcal{N}_{t_f}(i) = \{n+1\}$, as we connected the fusion token to all residues in the protein structure graph with a new type of edge $t_f$ (in both directions). To bridge the gap between the sequential and structure representation space, we utilize an

affine transformation $\text{Linear}_l$ (a linear layer) per each structure layer $l$ to project these representations into the same space. Subsequently, we set the fusion token representation $u_{n+1}^{(l)}$ as the input for the following sequential layer $l + 1$, resulting in:

$$z_{n+1}^{(l)} = \text{Linear}_{l'} \left( u_{n+1}^{(l)} \right)$$

where $\text{Linear}_{l'}$ is the affine transformation in the opposite direction (i.e., structure to sequential space or vice versa).

The protein representation undergoes transformations across the structure and sequential layers, where $L_1$ and $L_2$ are the number of sequential and structure layers, respectively, and $L_1 \geq L_2$ (note that $L_1$ need not be equal to $L_2$). Our aim is to uniformly integrate 3D structure information across the $L_1$ sequential and $L_2$ structure layers. Therefore, the fusion token is passed after $L_1 \div L_2$ sequential layers into the structure model. Then, the fusion token is passed back into the sequential model after every structure layer.

Lastly, the final hidden states of both the sequential and structure layers are concatenated to form the final protein output representation, which is:

$$h = \left[ z^{(L_1)}, u^{(L_2)} \right]$$

### 3.4 Pre-training Objective

Following GearNet (Zhang et al., 2023b) and ESM-GearNet (Zhang et al., 2023a), we use Multiview Contrastive learning as our pre-training objective, for a fair comparison. Other pre-training algorithms could also be applied. We compare different pre-training algorithms in Section 5.3.

For each protein structure graph $G$ (see Figure 2 for an illustration), we construct two augmented views, $g_x$ and $g_y$. We first perform subsequence cropping by sampling, for each view, a contiguous residue window $\mathcal{I} = [s, s+w]$ along the protein sequence and taking the induced subgraph $G[\mathcal{I}]$. This captures putative domains—recurring contiguous subsequences that often signify function (Zhang et al., 2023a). We then follow standard self-supervised learning practice by injecting stochastic noise in structure space: for each view, we perform random edge masking, independently dropping each remaining edge with probability $p = 0.15$. This generates structurally perturbed but still semantically consistent views of the same protein. Encoding $g_x$ and $g_y$ with FusionProt to get representations, projecting and $\ell_2$-normalizing them, we optimize an InfoNCE loss (van den Oord et al., 2018) where $(g_x, g_y)$ form the positive pair and negatives $g'$ are views from other proteins in the batch.

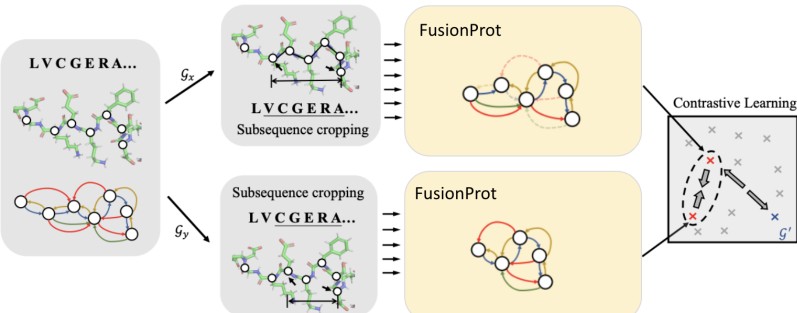

Figure 2: Illustration of the Multiview Contrastive pre-training objective (Zhang et al., 2023a). Two contiguous residue windows are cropped to form induced subgraphs $(g_x, g_y)$, 15% of edges are randomly masked, and our FusionProt's representations are aligned with InfoNCE against negatives $g'$ from other proteins in the batch.

## 4 Empirical Results

In this section, we present our empirical results. For each task, we report the mean and standard deviation of FusionProt's performance over five independent fine-tuning runs (with different random seeds) to ensure reproducibility. Additionally, we validate the statistical significance of performance differences, using a two-tailed paired t-test with a 95% confidence level ($p < 0.05$), comparing observations from the tested models. The normality of the paired differences was confirmed using the Shapiro–Wilk test (Shapiro & Wilk, 1965). Significant results are marked with an asterisk (*) in the tables.

Finally, we compute Cohen's d effect sizes (Cohen, 1969) to quantify the magnitude of FusionProt's improvements over baselines on each task. We observe large effects ($d > 0.8$) across all tasks, indicating that the improvements are not only statistically significant but also practically meaningful.

In Supplementary Section A.1, we outline our empirical setting, including the pre-training dataset, task details, baselines, implementation details, and a computational complexity analysis.

Table 1: Evaluation results on EC and GO prediction under various pre-trained baseline models. "PLM" and "Structure Info." indicate the usage of protein language models and structure information in the model, respectively. The $F_{max}$ score is the evaluation metric. Statistically significant results ($p < 0.05$) using a paired t-test across proteins in the test set are marked with an asterisk (*). For FusionProt, we report the mean and standard deviation over 5 independent fine-tuning runs (different seeds; EC: $n = 1604$, GO: $n = 3350$). The best result is highlighted in bold.

| Method | PLM | Structure Info. | EC | GO-BP | GO-MF | GO-CC |
|---|---|---|---|---|---|---|
| | | | $F_{max}$ | $F_{max}$ | $F_{max}$ | $F_{max}$ |
| ProtBERT-BFD (Elnaggar et al., 2021) | ✓ | ✗ | 0.838 | 0.279 | 0.456 | 0.408 |
| DeepFRI (Gligorijević et al., 2021) | ✓ | ✗ | 0.631 | 0.399 | 0.465 | 0.460 |
| ESM-1b (Rives et al., 2019) | ✓ | ✗ | 0.859 | 0.320 | 0.661 | 0.392 |
| ESM-2 (Lin et al., 2023) | ✓ | ✗ | 0.877 | 0.345 | 0.668 | 0.411 |
| GVP (Jing et al., 2021) | ✗ | ✓ | 0.886 | 0.495 | 0.672 | 0.420 |
| CDConv (Fan et al., 2023) | ✗ | ✓ | 0.820 | 0.453 | 0.654 | 0.479 |
| GearNet (Zhang et al., 2023b) | ✗ | ✓ | 0.871 | 0.481 | 0.650 | 0.476 |
| MIF-ST (Yang et al., 2022) | ✓ | ✓ | 0.803 | 0.239 | 0.627 | 0.322 |
| S-PLM (Wang et al., 2025) | ✓ | ✓ | 0.885 | 0.470 | 0.674 | 0.460 |
| ESM-GearNet (Zhang et al., 2023a) | ✓ | ✓ | 0.886 | 0.512 | 0.670 | 0.495 |
| SaProt (Su et al., 2024) | ✓ | ✓ | 0.884 | 0.486 | 0.678 | 0.479 |
| SaProt-GearNet (Su et al., 2024) | ✓ | ✓ | 0.886 | 0.512 | 0.672 | 0.504 |
| **FusionProt** | ✓ | ✓ | **0.904**\*±0.003 | **0.524**\*±0.004 | **0.689**\*±0.002 | **0.518**\*±0.004 |

### 4.1 Task 1: EC Number Prediction

In Table 1, we compared the performance of FusionProt with eleven baseline methods on the EC number prediction task. In particular, FusionProt significantly outperformed all baseline methods, achieving the highest $F_{max}$ score.

PLMs such as ProtBERT-BFD (Elnaggar et al., 2021) or ESM-2 (Lin et al., 2023), which rely solely on sequence data, produced substantially lower $F_{max}$ scores, compared to FusionProt. This highlights the benefit of incorporating 3D structure information into representation learning.

Similarly, FusionProt, which leverages both structure and sequential information, performed better than SOTA structure models such as GearNet (Zhang et al., 2023b) or CDConv (Fan et al., 2023), which trained only on 3D structure data. This indicates that while structure models are strong, they are not sufficient alone, as they require PLMs for optimal performance, probably due to their effective self-attention layers (Vaswani et al., 2017).

While models such as MIF-ST (Yang et al., 2022), S-PLM (Wang et al., 2025), ESM-GearNet (Zhang et al., 2023a), and SaProt-GearNet (Su et al., 2024) also attempt to utilize both types of information, they reduce one modality into a context for another model, leading to a loss of critical structure information. In addition, the importance of the fusion technique is emphasized by the significant underperformance of MIF-ST compared to other fusion models. Similarly to ESM-GearNet, MIF-ST uses outputs from a pre-trained sequence-only PLM as input to a structure model.

By integrating 1D and 3D protein structure information synergistically, FusionProt attempts to capture subtle structure features that influence enzyme specificity and activity, which are crucial for EC prediction. This performance improvement is essential for real-world applications, such as the diagnosis of enzyme deficiency-related diseases (Li et al., 2017).

## 4.2   Task 2: GO Term Prediction

Table 1 presents the results of the GO term prediction tasks compared to eleven baseline methods, including PLMs, structure models, and ensemble models. FusionProt showed strong performance across the board, achieving the highest $F_{\max}$ scores in all tasks, with a statistical significance.

The results demonstrate that sequential or structure information alone is insufficient, similarly to the EC prediction task (see Section 4.1). This indicates that the novel fusion of FusionProt is critical for achieving superior performance.

In the GO-BP task, which involves predicting a protein's role in biological processes, FusionProt achieves an $F_{\max}$ score of 0.524, significantly outperforming the next-best method. Biological processes often rely on long-range structural interactions and cooperative protein functions, which makes this task particularly sensitive to accurate 3D structure representations. The substantial performance gap between SaProt and SaProt-GearNet further emphasizes the critical role of structure in this setting, a factor that FusionProt leverages more effectively through early integration.

In the GO-CC task, which predicts the cellular component of proteins, FusionProt achieves an $F_{\max}$ of 0.518, outperforming all baselines including ESM-GearNet (Zhang et al., 2023a), probably due to the more direct connection between 3D structure characteristics and subcellular localization (Gillani & Pollastri, 2024). The prediction of cellular components is strongly based on the spatial organization of proteins within the cell, which is well represented by detailed 3D structures (Song et al., 2022).

This reliance on structural information benefits models such as CDConv (Fan et al., 2023), although it exhibits weaker performance in EC prediction. In contrast, predicting molecular functions involves complex interactions and dynamic changes that are less directly captured by 3D structures (Saraç et al., 2010), resulting in a comparatively smaller performance gap than in the other tasks.

## 4.3   Task 3: Mutation Stability Prediction

Table 2 reports the results for the Mutation stability prediction (MSP) task, which evaluates a model's ability to assess the effect of amino acid substitutions on protein stability, a critical problem in protein engineering, variant effect prediction, and drug design.

Following the evaluation protocol of ESM-GearNet (Zhang et al., 2023a), we compare FusionProt on the MSP task against the SOTA structure-based method for this task, GVP (Jing et al., 2021) and the joint sequence–structure baseline ESM-GearNet (Zhang et al., 2023a). Sequence-only PLMs do not take structural inputs and therefore cannot address structure-dependent tasks such as MSP (Zhang et al., 2023a).

FusionProt achieves the highest AUROC among all evaluated methods, reaching 0.745 with statistical significance ($p < 0.05$), outperforming both structure-aware (GVP) and sequence-structure fused (ESM-GearNet) baselines. Notably, FusionProt improves upon ESM-GearNet by approximately 24.37%, and surpasses the current SOTA GVP by 5.1%. This highlights the effectiveness of our iterative fusion mechanism in preserving and integrating long-range sequence and structural dependencies, which are especially important for modeling stability changes that may arise from distal or context-dependent mutations.

Compared to GVP, which is limited to local spatial interactions, and ESM-GearNet, which performs shallow one-shot fusion, FusionProt allows multi-layer cross-modal refinement via a learnable token and bidirectional interaction path. This enables a more expressive and biologically grounded representation, particularly for capturing distributed compensatory effects in allosteric regions, commonly overlooked in residue-centric models.

Table 2: Evaluation results on MSP prediction across different models. "PLM" and "Structure" indicate the usage of protein language models and structure information in the model, respectively. The AUROC score is used as the evaluation metric. Statistically significant results ($p < 0.05$) using a paired t-test across proteins in the test set are marked with an asterisk (*). For FusionProt, we report the mean and standard deviation over 5 independent fine-tuning runs (different seeds; MSP: $n = 347$). The best result is highlighted in bold.

| Method | PLM | Structure | MSP (AUROC) |
|---|---|---|---|
| ESM-GearNet (Zhang et al., 2023a) | ✓ | ✓ | 0.599 |
| GVP (Jing et al., 2021) (SOTA) | ✗ | ✓ | 0.709 |
| **FusionProt** | ✓ | ✓ | $\mathbf{0.745}^*\pm0.006$ |

# 5 Ablation and Analysis

We conduct a series of ablation and diagnostic studies to evaluate the key design decisions behind FusionProt. These include varying the fusion-injection frequency, comparing different structure encoders, and testing alternative pre-training objectives. We also assess the model's robustness to noise in predicted 3D structures. Finally, we present biological case studies that highlight FusionProt's practical utility in real-world tasks such as drug discovery and disease research.

## 5.1 Ablation on Fusion-Injection Frequency

We aimed to determine the optimal number of fusion injections in our FusionProt model, which fused information between the structure and sequential models using a specialized learnable fusion token. In our standard method, given a sequential model with $L_1$ layers and a structure model with $L_2$ layers, where $L_1 \geq L_2$, the fusion token is passed after $L_1 \div L_2$ sequential layers into the structure model (see Section 3.3.2). Then, the fusion token is passed back into the sequential model after every structure layer. In this ablation, we experimented with different injection frequencies: half the regular number (every $2 \cdot (L_1 \div L_2)$ sequential layers and two structure layers) and one-third the regular number (every $3 \cdot (L_1 \div L_2)$ sequential layers and three structure layers). The results in Table 3 indicate that the standard fusion injection frequency consistently provides the best performance across tasks. Reducing fusion injections negatively impacted performance, highlighting the importance of the fusion token. By decreasing the frequency of injections, the model's ability to integrate these two types of information is compromised, leading to suboptimal predictions.

Table 3: Ablation test for the number of fusion injections in FusionProt. Statistically significant results with $p < 0.05$ using a t-test are marked with an asterisk (*). The best result is highlighted in bold.

| Number of Fusion | EC | GO-BP | GO-MF | GO-CC |
|---|---|---|---|---|
| | $F_{\max}$ | $F_{\max}$ | $F_{\max}$ | $F_{\max}$ |
| Few (One-Third) | 0.876 | 0.511 | 0.659 | 0.498 |
| Medium (Half) | 0.882 | 0.514 | 0.669 | 0.504 |
| Full (Standard) | $\mathbf{0.904}^*$ | $\mathbf{0.524}^*$ | $\mathbf{0.689}^*$ | $\mathbf{0.518}^*$ |

## 5.2 3D Structure Models Comparison

FusionProt is designed to be flexible and modular, allowing the structure encoder component to be replaced with alternative architectures. The ablation study for various SOTA 3D structure models is detailed in Table 4. We evaluated several structure models, including GearNet (Zhang et al., 2023b), GVP (Jing et al., 2021), and CDConv (Fan et al., 2023). For each structure model, we applied our fusion technique and compared its performance with FusionProt, which utilizes the GearNet (Zhang et al., 2023b) model (see Section 3.2.3). The GVP model replaces standard MLPs (Murtagh, 1991) in GNN (Scarselli et al., 2009) layers with generalized vector perceptrons, which handle scalar and geometric features as vectors that adapt to spatial rotations. In contrast, CDConv utilizes GearNet's multi-type message passing to capture sequential and spatial interactions among residues. For consistency, we follow the model selection protocol of ESM-GearNet (Zhang et al., 2023a). Our results demonstrate that the FusionProt model, utilizing the GearNet structure, consistently achieves superior performance compared to other models. This aligns with previous studies (Zhang et al., 2023a), which have also established GearNet as a SOTA approach for joint representation learning.

Table 4: Ablation test for various structure models used by FusionProt. For all models, the PLM is the ESM-2 (Lin et al., 2023). Statistically significant results with $p < 0.05$ using a t-test are marked with an asterisk (*). The best result is highlighted in bold.

| Method | EC | GO-BP | GO-MF | GO-CC |
|---|---|---|---|---|
| | $F_{\max}$ | $F_{\max}$ | $F_{\max}$ | $F_{\max}$ |
| FusionProt (GVP) | 0.889 | 0.507 | 0.677 | 0.462 |
| FusionProt (CDConv) | 0.837 | 0.478 | 0.665 | 0.497 |
| FusionProt (GearNet) | **0.904**$^*$ | **0.524**$^*$ | **0.689**$^*$ | **0.518**$^*$ |

## 5.3 Pre-training Algorithm Ablation

Table 5 compares FusionProt, trained with Multiview Contrastive learning (Section 3.4), with other pre-training algorithms. First, we tested self-prediction methods (Zhang et al., 2023b), which aim to predict one part of the protein given the remaining context. Specifically, we utilized the Residue Type Prediction method, a self-supervised task that performs masked prediction on individual residues. Next, we evaluated diffusion-based methods (Zhang et al., 2023c), inspired by the success of diffusion models in capturing the relationship between sequences and structures. During training, noise levels are added to structures and sequences, with higher noise levels indicating more distortion. We tested the SiamDiff objective, which refines structures via torsional adjustments and noise reduction. The results show that the Multiview Contrastive approach outperformed both Residue Type Prediction (using only sequence data) and SiamDiff (which integrates sequence and structure separately). Unlike methods that treat sequence and structure separately, Multiview Contrast integrates both, aligning subsequence representations from the same protein to capture interrelated sequences and structure patterns.

Table 5: Ablation test for the FusionProt model with different pre-training algorithms. Statistically significant results with $p < 0.05$ using a t-test are marked with an asterisk (*). The best result is highlighted in bold.

| Method | EC | GO-BP | GO-MF | GO-CC |
|---|---|---|---|---|
| | $F_{\max}$ | $F_{\max}$ | $F_{\max}$ | $F_{\max}$ |
| Residue Type | 0.890 | 0.524 | 0.673 | 0.506 |
| SiamDiff | 0.874 | 0.509 | 0.645 | 0.512 |
| Multiview Contrast | **0.904**$^*$ | **0.524**$^*$ | **0.689**$^*$ | **0.518**$^*$ |

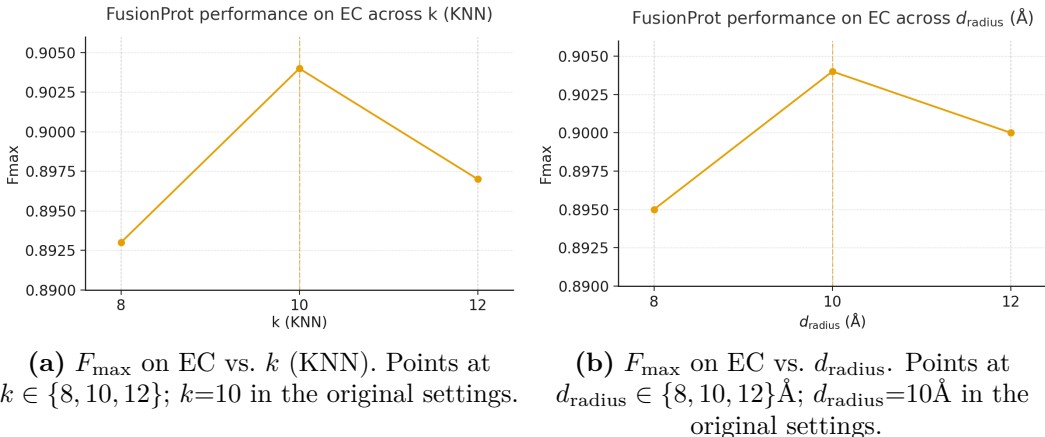

**(a)** $F_{\max}$ on EC vs. $k$ (KNN). Points at $k \in \{8, 10, 12\}$; $k$=10 in the original settings.

**(b)** $F_{\max}$ on EC vs. $d_{\text{radius}}$. Points at $d_{\text{radius}} \in \{8, 10, 12\}$Å; $d_{\text{radius}}$=10Å in the original settings.

Figure 3: Ablation of graph-construction parameters on the EC prediction task. $F_{\max}$ as a function of neighborhood size: $k$ (KNN) and $d_{\text{radius}}$ (spatial threshold). All other settings are fixed.

### 5.4 Robustness to Low-Confidence Regions in 3D Structures

Robustness in protein modeling often refers to a model's ability to tolerate minor perturbations (Cho et al., 2024)—an important property given that predicted structures (e.g., from AlphaFold2) may contain low-confidence regions, reflected as segments with low pLDDT scores. These regions are less reliable for practical use. To assess FusionProt's robustness in such scenarios, we simulate low-confidence regions by injecting Gaussian perturbations into the alpha-carbon backbone coordinates of AlphaFold2-predicted structures. The noise was zero-mean, with standard deviations ranging from 0.1 Å to 1.5 Å, simulating local structure deviations commonly observed in predicted or low-confidence regions. We then fine-tune the model on the EC and GO-BP downstream tasks and evaluate performance across these noise levels.

FusionProt maintained stable performance up to a noise level of 0.9 Å, with only a modest 3–5 points absolute drop in $F_{\max}$ on EC and GO-BP tasks beyond this threshold. These results indicate that FusionProt is resilient to realistic structural noise and remains effective even when relying on imperfect 3D structures, as commonly encountered in high-throughput or large-scale prediction settings.

### 5.5 Graph Construction Parameters Ablation

We ablate graph-construction choices on the EC prediction task (see Figure 3) by varying $k$ (of the KNN) and the spatial threshold $d_{\text{radius}}$ used to build the protein structure graph (see Section 3.2.2). Across $k \in \{8, 10, 12\}$ and $d_{\text{radius}} \in \{8, 10, 12\}$ Å, performance is stable with a broad optimum near $k$=10 and $d_{\text{radius}}$=10 Å, indicating that FusionProt is robust to reasonable graph choices and that our defaults are optimal and recommended.

Across EC and GO tasks, FusionProt is largely robust to protein length, which we attribute chiefly to our sequence-truncation protocol (see Section A.1.4). When stratifying input proteins by sequence length, longer proteins ($\gtrsim 600$ residues) benefit slightly from denser connectivity ($k$=12 or $d_{\text{radius}}$=12 Å), which better preserves long-range contacts, resulting in higher $F_{\max}$ in downstream tasks (EC: +0.5 percentage points in $F_{\max}$ with $p < 0.05$ using a t-test).

### 5.6 Biological Case Studies

FusionProt consistently outperforms SOTA baselines across all downstream tasks (Table 1, Table 2). When such performance gains are driven by biologically meaningful inputs, such as detailed 3D structural features, they can indicate that the model is capturing relationships of functional relevance.

To better understand the nature of these improved learned protein representations, we evaluated a model's ability to predict EC numbers, as outlined in Section A.1.2, using pre-trained representations without any

fine-tuning. This approach enables an assessment of the inherent quality of the learned embeddings, independent of task-specific training. We then compared FusionProt's predictions with those of ESM-GearNet, focusing on the cases with the largest discrepancies.

In Supplementary Table 6, we present these proteins into distinct biological groups based on shared mechanistic characteristics. We focus on two common mechanisms and provide a representative example for each group (see Supplementary Section A.2).

## 6 Conclusions

In this paper, we introduced FusionProt, a novel architecture designed to learn a unified representation of the 1D and 3D structures of proteins simultaneously. FusionProt incorporates a specialized learnable fusion token that enables an iterative exchange of information between a PLM and the protein's 3D structure graph, facilitating a more comprehensive representation through iterative learning cycles between a PLM and a structure model.

We propose a novel fusion algorithm that enables effective propagation of information between a PLM and a structure model. This fusion technique outperforms previous methods, which often convert one modality into context for another model, potentially leading to the loss of crucial structural information.

We evaluated our proposed method on several protein-level tasks to assess its effectiveness in protein representation learning. These tasks include protein annotations, which are essential for real-world medical applications. FusionProt significantly outperformed all baseline methods, including SOTA models, across all tasks, with statistically significant improvements.

We perform ablation tests to examine the contribution of our novel approach of learning unified representations of both 1D and 3D structures of proteins simultaneously. The tests confirm that our approach significantly enhances FusionProt's ability to capture the intricate relationships between protein sequences and their 3D structures, leading to improved performance across various real-world tasks.

Although the proposed fusion token mechanism is tailored for bidirectional interaction between sequence and structure modalities, it is conceptually extensible to more than two information channels. For example, additional modalities such as ligand descriptors, protein dynamics, or expression context could, in principle, be integrated into the framework by assigning separate fusion tokens or adopting a shared token passed through each encoder block in sequence. We leave the systematic evaluation of such multi-modal extensions to future work. Additionally, although we evaluate MSP for residue-level perturbations, MSP does not replace standardized DMS benchmarks; it is closely related and indicates benefits from structural cues. In future work, we will expand evaluation to DMS-based datasets (e.g., ProteinGym).

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

# A Appendix

## A.1 Empirical Evaluation

In this section, we outline our empirical setting. We performed analysis on several downstream tasks, demonstrating the value of the representation we provide in real-world bioinformatics tasks.

### A.1.1 Protein Structure Dataset

To reduce reliance on the availability and quality of external structural data, we leverage AlphaFold2 (Jumper et al., 2021), a well-known SOTA model, which enables us to generate 3D structures and eliminate the need for external sources. Furthermore, many recent high-performing methods, such as SaProt (Su et al., 2024) and ESM-GearNet (Zhang et al., 2023a), also used it for structure generation, ensuring consistency in comparisons. Therefore, we use the AlphaFold protein structure database (Váradi et al., 2023; 2021) for pre-training, which contains 805K protein structures predicted by AlphaFold2 (Jumper et al., 2021). Alternatively, other models, such as ESMFold (Lin et al., 2022), can be applied. Since AlphaFold predictions may include local inaccuracies, we assess FusionProt's robustness to structure noise in Section 5.4.

### A.1.2 Tasks

We evaluated our proposed method on several common downstream tasks, selected based on SOTA works of GearNet (Zhang et al., 2023a) and SaProt (Su et al., 2024), to assess its effectiveness for protein representation learning. These tasks include protein annotations such as Enzyme Commission (EC) prediction, Gene Ontology (GO) prediction, and Mutation Stability Prediction. The GO prediction task includes three sub-tasks: predicting a protein's biological processes (BP), molecular functions (MF), and cellular components (CC).

To ensure the validity of our results, all downstream task splits are constructed so that sequences in one set share no more than 30% Needleman–Wunsch sequence identity with any sequence in the other sets. In addition, the hold-out sets for each task share no more than 30% identity with any protein in the AlphaFold2 pre-training corpus (Jumper et al., 2021).

**Enzyme Commission Number Prediction** Annotation of enzyme function has a wide range of real-world applications, including metagenomics, diagnosis of enzyme-deficiency-related diseases (Li et al., 2017), and cellular metabolism (Ryu et al., 2019). This task focuses on determining enzyme function by predicting EC numbers, which characterize a protein's catalytic activity in biochemical reactions. It involves 538 binary classification problems derived from the third and fourth levels of the EC classification tree (Webb, 1992). We used dataset splits from DeepFRI (Gligorijević et al., 2021), and the evaluation metric is the $F_{\max}$ score.

**Gene Ontology Term Prediction**  The GO knowledgebase (Harris et al., 2004) provides a set of structured and controlled terms describing gene products and their molecular properties. Many real-world biological applications, such as predictions of protein-protein interactions, rely on GO term-protein annotations (Pesquita et al., 2007; Jiang & Conrath, 1997). This benchmark includes three tasks: predicting a protein's biological processes, molecular functions, and cellular components. Each task involves multiple binary classification problems based on GO term annotations. We used dataset splits from DeepFRI (Gligorijević et al., 2021), and the evaluation metric is the $F_{\max}$ score.

**Mutation Stability Prediction**  Mutation stability prediction (MSP) is crucial in bioinformatics for understanding how genetic mutations affect protein stability, which plays a key role in disease mechanisms and drug development. This task aims to predict whether a mutation enhances a protein complex's stability. We utilize the datasets and hyperparameters from ESM-GearNet (Zhang et al., 2023a). Evaluation is based on AUROC.

### A.1.3  Baselines

We compare with numerous baseline models, including PLMs and structure models. We used ProtBERT-BFD (Elnaggar et al., 2021), DeepFRI (Gligorijević et al., 2021), ESM-1b (Rives et al., 2019), and ESM-2 (Lin et al., 2023) as SOTA PLM sequential models. GearNet (Zhang et al., 2023b), GVP (Jing et al., 2021), and CDConv (Fan et al., 2023) are used as SOTA structure models. Furthermore, we include MIF-ST (Yang et al., 2022), S-PLM (Wang et al., 2025), ESM-GearNet (Zhang et al., 2023a), SaProt (Su et al., 2024) and SaProt-GearNet (Su et al., 2024) as SOTA in joint learning of structure models with sequential models. For consistency, when using ESM-2 we used the ESM-2-650M variant (Lin et al., 2023). In addition, GearNet (Zhang et al., 2023b), ESM-GearNet (Zhang et al., 2023a), and FusionProt are pre-trained with the same objective, which is the Multiview Contrast (Zhang et al., 2023b) objective, as prior work showed its superior performance.

### A.1.4  Implementation Details

**Pre-Training Phase**  We follow ESM-GearNet (Zhang et al., 2023a) in adopting the same training objective and model selection strategy. Our model uses a pre-trained ESM-2-650M (Lin et al., 2023) as the base PLM, with 33 layers ($L_1 = 33$), and GearNet (Zhang et al., 2023b) with 6 layers ($L_2 = 6$) and 512 hidden dimensions as the structure encoder. The embedding dimension $D$ is set to 1280. Multiview Contrast (Zhang et al., 2023b) is used as the pre-training objective. Hyperparameters were tuned using the same search procedure and ranges reported in ESM-2 (Lin et al., 2023) and ESM-GearNet (Zhang et al., 2023a), with the best configuration selected based on validation performance. For FusionProt, this configuration corresponded to training for 50 epochs with a learning rate of 2e-4 and a global batch size of 256 proteins. To accommodate long sequences, inputs are truncated to a maximum of 1,024 tokens. All implementations use the TorchDrug library.

**Fine-Tuning Phase**  During inference, we incorporate task-specific classification heads to generate predictions for each downstream task. Following the protocol of the recent SOTA model SaProt (Su et al., 2024), we evaluated our model and baselines under a consistent hyperparameter tuning procedure to ensure fair comparison. For all methods, we performed tuning within the hyperparameter ranges reported in SaProt (Su et al., 2024), with the best configuration for each model selected based on validation performance. We use AdamW with $\beta_1 = 0.9$ and $\beta_2 = 0.98$, treating learning rate, weight decay, and batch size as tunable; default initial values and exact ranges are provided in our GitHub repository. All models were trained to convergence, and the final checkpoint was chosen by the highest validation score (task-specific primary metric).

### A.1.5  Computational Complexity

FusionProt was trained on 4× NVIDIA A100 80 GB GPUs for 48 hours (192 GPU-hours). It retains the underlying architecture of the base sequential and structure encoders (e.g., ESM-2 and GearNet), applying each layer once as in prior baselines (e.g., ESM–GearNet).

Let $L$ denote the protein sequence length (in our settings, we limit the input to $L = 1{,}024$). FusionProt adds one token on the sequential encoder and one universally connected node on the structure encoder. On the sequential side, per layer the self-attention size changes from $L^2$ to $(L+1)^2$, a relative increase of $\frac{(L+1)^2 - L^2}{L^2} = \frac{2}{L} + \frac{1}{L^2}$; at $L = 1{,}024$ this is $\approx 0.195\%$ per layer. Because the baseline and FusionProt use the same depth, this fractional increase does not amplify with depth; in practice, feed-forward blocks dominate compute, so the impact is small.

On the structure side, the fusion node contributes $L$ additional edges of a new type. If the original graph has on average $M$ edges per residue, the baseline has $\approx ML$ edges; adding $L$ fusion edges increases message passing by about $1/M$. With an approximate bound of $M=100$ edges per residue in our settings, this is a $\sim 1\%$ overhead per structure layer. Since GearNet has 6 layers and message passing is a minority of end-to-end cost, the overall runtime and memory impact remain modest.

Empirically, we observe a consistent 2–5% increase in inference latency and throughput across sequence lengths and model scales; for example, latency rises from $12.0\,\text{ms}$ to $12.6\,\text{ms}$ per 1,000 residues at $L{=}1{,}024$ ($\sim 5\%$). We also observe a small peak-memory increase ($\approx 2.6\%$) from the extra token activations and fusion edges, and about a 3% increase in training step time with the same number of epochs to convergence. Together, the scaling analysis and these measurements support that FusionProt introduces a small, well-bounded overhead.

## A.2 Biological Insights

FusionProt consistently outperforms SOTA baselines across all downstream tasks (Table 1, Table 2). When such performance gains are driven by biologically meaningful inputs, such as detailed 3D structural features, they can indicate that the model is capturing relationships of functional relevance. However, we note that one must avoid conflating improved prediction with causation: a model's ability to predict biological outcomes with more precision does not in itself support causal inferences. Nevertheless, significant increases in specific classes of predictions can provide valuable hints that frame hypotheses, guide experimental questions, and motivate follow-up studies.

For FusionProt, the largest improvements are often observed in proteins whose function, stability, or interactions depend strongly on complex structural organization or long-range spatial effects. In contrast to methods such as ESM-GearNet (Zhang et al., 2023a) or SaProt (Su et al., 2024), which integrate sequence and structure in a single pass, FusionProt employs an iterative fusion token mechanism that enables multiple rounds of bidirectional information exchange between sequence and structure encoders. This repeated refinement allows the model to amplify subtle, spatially localized features that might otherwise be diluted in one-shot fusion.

To better understand the nature of these improved learned protein representations, we evaluated a model's ability to predict EC numbers, as outlined in Section A.1.2, using pre-trained representations without any fine-tuning. This approach enables an assessment of the inherent quality of the learned embeddings, independent of task-specific training. We then compared FusionProt's predictions with those of ESM-GearNet, focusing on the cases with the largest discrepancies.

We then organized these proteins into distinct biological groups based on shared mechanistic characteristics. We focus on two common mechanisms (Table 6) and provide a representative example for each group.

**Structural subunits for macromolecular assembly.** The bacterial RNA polymerase $\omega$ subunit (Figure 4) is small and poorly conserved, yet plays a key structural role in holoenzyme assembly by recruiting and stabilizing the $\beta'$ subunit (Mathew & Chatterji, 2006). Its short length and weak sequence signal make it difficult for sequence-only models to classify correctly. FusionProt's iterative sequence–structure fusion appears to preserve features on the $\beta'$ contact surface that are tied to assembly and stability, while one-pass fusion may reduce such localized signals. Consistent with this interpretation, FusionProt assigned the correct complex-level EC labels with high confidence (mean of 0.90), while the baseline ESM-GearNet (Zhang et al., 2023a) did not (mean of 0.21). Because $\omega$ is noncatalytic and the EC number reflects holoenzyme activity, we hypothesize that FusionProt recognizes the context of the quaternary structure, specifically an

Table 6: Mechanisms highlighted by our case studies, with generalizable structural signals, a representative example, and EC-probe confidence.

| Biological mechanism | Insight / structural signal | Example protein | Confidence improvement |
|---|---|---|---|
| Assembly interface | Interface fingerprint at subunit contacts (shape/electrostatics/hydrophobicity); gains when function depends on quaternary context rather than active-site chemistry. | RNAP $\omega$–$\beta'$ interface | 0.90 vs 0.21 |
| Loop-gated pocket | Dynamic loops that gate ligand/catalytic sites; loop residues occupy broader allowed $(\phi, \psi)$ basins (flexibility) supporting access and specificity. | D-Ala–D-Ala ligase (ATP-grasp) | 0.88 vs 0.62 |

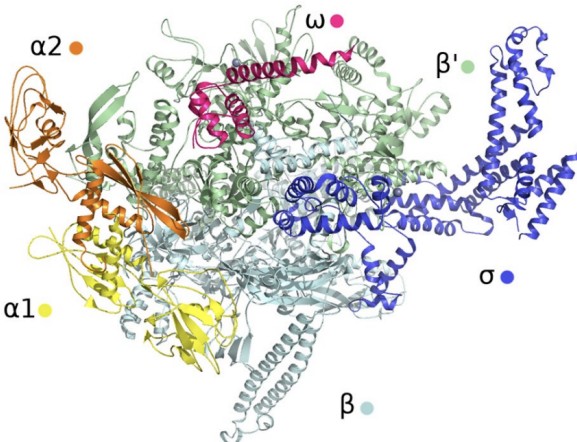

Figure 4: DNA-directed RNA polymerase highlighting the $\omega$–$\beta'$ assembly interface ($\omega$ in magenta, $\beta'$ in light green) (Kurkela et al., 2021). Although $\omega$ is small and poorly conserved, it is essential for recruiting and stabilizing $\beta'$ during holoenzyme assembly. FusionProt captured this interface-specific structural fingerprint and classified the correct complex EC labels with high confidence, while ESM-GearNet failed.

interface-centric fingerprint of local shape, charge, and hydrophobic patterning at the $\omega$–$\beta'$ surface, rather than active-site chemistry. Figure 4 highlights this interface (magenta: $\omega$; light green: $\beta'$), where the model appears to localize signal more strongly than the baseline. This implies that weakening these interface features would reduce assembly propensity and lower model confidence, providing a concrete avenue for validation.

**Conserved ATP-dependent peptide ligases.** D-alanine–D-alanine ligase (Figure 5) is a well-characterized member of the ATPgrasp enzyme superfamily, which catalyzes the formation of the D-Ala-D-Ala dipeptide, an essential step in bacterial peptidoglycan biosynthesis. Its conserved 3D fold encloses an ATP-binding site and supports relatively simple, well-understood catalytic chemistry. Both FusionProt and ESM-GearNet correctly predicted its EC classes, but FusionProt did so with higher confidence (mean of 0.88 vs. 0.62). We hypothesize that the iterative fusion mechanism amplifies local structural cues from the mobile loops surrounding the ligand-binding pocket, which mediate substrate recognition and catalysis, beyond what the global fold alone conveys. Consistent with this, the Ramachandran plot (Saleem et al., 2021) for the same structure (Figure 5) shows a broader distribution of the dihedral angles ($\phi$ and $\psi$ angles) of the backbone for the loop residues, within allowed regions: hallmarks of conformational flexibility such as pocket gating and substrate specificity. This suggests that repeated sequence–structure refinement can

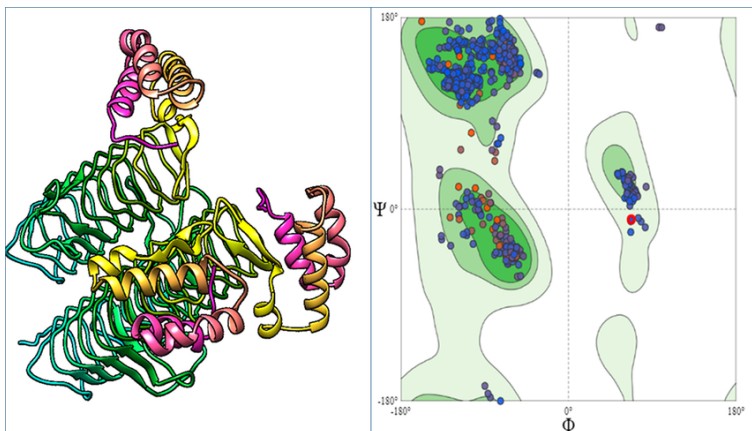

Figure 5: D-alanine–D-alanine ligase. Left: 3D structure showing the ATP-binding cleft and the surrounding pocket-adjacent loops. Right: Ramachandran plot (Saleem et al., 2021) of the dihedral angles ($\phi$ and $\psi$ angles) of backbone for the same protein; shaded contours denote sterically allowed conformations. The broader spread within allowed basins is characteristic of flexible loop regions that gate the pocket, supporting our interpretation that FusionProt captures this loop-centric fingerprint and yields higher EC prediction confidence than ESM-GearNet.

sharpen flexible, function-critical motifs even in enzymes with highly conserved catalytic cores, implying that perturbations to these loops could influence both enzymatic activity and model confidence.

Across categories, FusionProt's unified embeddings capture structural determinants often missed in less iterative fusion frameworks, such as allosteric loops, oligomerization interfaces, and convergent active-site geometries. These enriched representations may facilitate advances in function prediction, mutagenesis design, drug discovery, and the annotation of orphan proteins from metagenomic datasets.

### A.3 Interpretability Analyses

We examine how the fusion token aggregates structural cues during EC prediction. FusionProt augments the sequence encoder with a single fusion token and periodically injects structure-derived information into the sequence stack (every few Transformer layers). To visualize what the token attends to, at each layer we record the attention distribution from the fusion token to all sequence positions (averaged over heads). For plotting, we remove padding and special tokens, and we normalize each layer's fusion-to-residue row over residues so patterns remain visible for long sequences. We also track the fusion token's self-attention (fusion to fusion) on its original softmax scale. Horizontal guidelines mark structure-injection depths (every 5 layers).

Figure 6 shows three representative proteins of different lengths. In each panel, the heatmap (left) depicts fusion token to residue attention across depth, and the thin strip (right) shows the token's self-attention per layer. Across cases we observe: (i) attention progressively concentrates on a small subset of residues in later layers, indicating targeted querying rather than diffuse patterns; (ii) immediately after each structure injection, the attention map refocuses, consistent with incorporation of new 3D evidence; and (iii) self-attention varies with depth and is not persistently high, suggesting the token predominantly routes attention outward to residues. Together, these observations indicate that FusionProt integrates structural cues into the sequence stream and contributes an informative signal rather than noise.

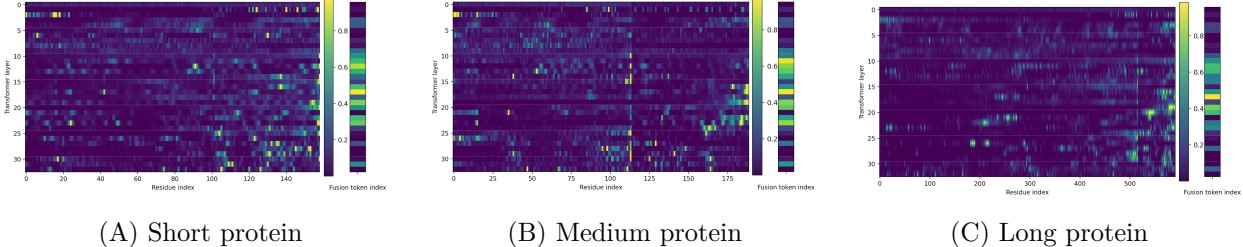

(A) Short protein        (B) Medium protein        (C) Long protein

Figure 6: Attention patterns of the fusion token on the EC prediction task. In each panel, the heatmap on the left shows fusion-token to residue attention across transformer layers (rows) and residue indices (columns). Padding and special tokens are removed; each row is normalized over residues to aid visibility. The thin strip on the right shows fusion-token self-attention per layer on the original softmax scale. Horizontal guidelines mark the structure-injection depths. Across short (A), medium (B), and long (C) proteins, attention sharpens in deeper layers and refocuses after injections, while self-attention remains non-saturated, indicating that the fusion token queries specific residue subsets rather than self-attending.

