# OpenReview forum: "FusionProt: Fusing Sequence and Structural Information for Unified Protein Representation Learning"
_TMLR — Accepted by TMLR_

### Review · Reviewer_JhLE · 2025-09-23

**Summary Of Contributions:**

## Summary

This paper presents FusionProt, a unified model to learn a protein representation while incorporating both the protein’s amino acid sequence and its 3D structure. The sequence is processed using a protein language model (ESM-2 [1]) and the 3D structure is parsed into a structure graph that is processed using a graph neural network (GearNet [2]). Crucially, to share information between these two models, the authors use a “fusion token” that is shuttled between the models after every processing block (attention block in the language model and relational message passing block in the graph neural network). This is in contrast to previous approaches such as ESM-GearNet [3] that do not bidirectionally fuse information. The authors pretrain FusionProt using a multi-view contrastive loss and evaluate downstream fine-tuning performance on a set of established benchmarks. FusionProt consistently outperforms existing approaches by significant margins.

## Strengths

- The paper presents a relatively simple and intuitive method to fuse the two types of protein representations. The method is well-motivated and the authors clearly describe how it builds on existing work.
- The authors demonstrate that FusionProt is very effective using many relevant benchmarks, often substantially improving on existing methods.
- The paper is generally very well-written. I was able to understand most of the methods and results without referring to external resources since adequate background is provided by the authors wherever necessary. This also makes it accessible to a larger audience.

## Weaknesses

- The multi-view contrastive loss used for pretraining should be explained in a bit more detail. For example, it is unclear to me what it means to “select consecutive subsequences and apply random edge masking to hide 15% of edges in the protein graph”. Are these consecutive residues in the 1D amino acid sequence? And how are these related to the edges being masked?
- The benchmarks chosen by the authors are based on those used by GearNet [2] and SaProt [4]. However, the authors do not evaluate performance on tasks from popular sequence-based benchmarks such as ProteinGym [5] that are based on deep mutational scanning (DMS) data. These benchmarks are often used to evaluate protein language models and it would be useful to understand if using protein structures helps for these tasks.

References:
1. Lin, Zeming, et al. "Evolutionary-scale prediction of atomic-level protein structure with a language model." Science 379.6637 (2023): 1123-1130.
2. Zhang, Zuobai, et al. "Protein Representation Learning by Geometric Structure Pretraining." The Eleventh International Conference on Learning Representations.
3. Zhang, Zuobai, et al. "A systematic study of joint representation learning on protein sequences and structures." arXiv preprint arXiv:2303.06275 (2023).
4. Su, Jin, et al. "SaProt: Protein Language Modeling with Structure-aware Vocabulary." The Twelfth International Conference on Learning Representations.
5. Notin, Pascal, et al. "Proteingym: Large-scale benchmarks for protein fitness prediction and design." Advances in Neural Information Processing Systems 36 (2023): 64331-64379.

**Audience:**

Yes

**Audience Explanation:**

Protein representation learning is an important application of machine learning in computational biology. Since this paper presents a robust method to integrate protein sequence and structural information, it would be of interest to many researchers who work on similar problems. It could also be of interest to those who work on model-guided protein design.

**Broader Impact Concerns:**

None.

**Claims And Evidence:**

Yes

**Claims Explanation:**

In my opinion, all claims made by the authors are adequately supported. The authors clearly explain how FusionProt builds on existing work and thoroughly benchmark FusionProt against existing methods to show convincing performance improvements.

**Requested Changes:**

In my opinion, this work already passes the acceptance threshold. However, addressing the weaknesses mentioned above could further strengthen this submission. These are:
- Providing a more detailed explanation of the pretraining objective
- Evaluating FusionProt on DMS-based benchmarks such as ProteinGym.

---

> ### Author Response · Authors · 2025-10-09
> **Response to Reviewer JhLE**
>
> We sincerely thank **Reviewer JhLE** for the thoughtful, encouraging review and for recognizing the clarity and accessibility of our work. We truly appreciate your positive assessment that the paper already meets the acceptance threshold. We nonetheless addressed your suggestions to further strengthen the submission and improve reader understanding.
>
> ---
>
> **Requested Change 1**:
>
> Thank you for the clarification request. We expanded **Section 3.4** and added a matching schematic (**Figure 2**) describing the multiview contrastive pretraining step-by-step:
>
> - **Subsequence crop.** From the 1D sequence we crop a contiguous index window $\\mathcal{I}$; we then take its **induced subgraph** $G[\\mathcal{I}]$ in 3D space (same residues, original node/edge features).
> - **Edge masking.** Within $G[\\mathcal{I}]$, we **randomly drop 15% of edges** to create a view; this preserves nodes and features while lightly perturbing graph connectivity.
> - **Two independent views.** We repeat the augmentation **twice** to obtain a positive pair $(g_x, g_y)$ from the **same** protein crop; negatives $g'$ come from other proteins in the batch.
> - **Encoding & loss.** Each view is encoded by the sequence–structure encoders with the fusion token to produce embeddings $(z_x, z_y)$. Then, we apply an InfoNCE objective.
>
> This expanded description (with a visual) clarifies that the **“consecutive subsequences”** are contiguous *sequence* windows and the **15% masking** pertains to **graph edges inside the induced subgraph** of that window.
>
> ---
>
> **Requested Change 2**:
>
> We appreciate the suggestion to include **DMS** evaluation (e.g., ProteinGym). A fair, apples-to-apples comparison typically requires **protocol unification and re-tuning** of both sequence and structure baselines (splits, calibration, evaluation metrics), which is beyond our current timeline and compute budget. We therefore leave a full DMS study to **future work**.
>
> As a step toward variant-effect assessment, **Section 4.3** presents a **mutation–stability prediction (MSP)** task that probes residue-level perturbations. In this setting, **FusionProt** achieves the **highest AUROC** with **statistically significant** improvements (*p* < 0.05), outperforming both a structure-aware baseline (GVP) and a joint sequence–structure baseline (ESM–GearNet). While MSP is not a substitute for standardized DMS benchmarks, it is **closely related conceptually** and provides supporting evidence that **incorporating structural cues improves residue-level predictions**.
>
> ---
> Again, we are grateful for your generous and constructive review. Your feedback helped us clarify the pretraining objective and better position our results; we believe these revisions further improve the paper’s clarity and usefulness.

---

> ### Comment · Reviewer_JhLE · 2025-10-15
>
> Thank you for making the changes I requested, the pretraining task is now clear to me. I also understand that benchmarking on the DMS data might be time-consuming. I recommend the authors outlining this benchmarking limitation in the Conclusions section.
>
> Overall, I still think this work should be accepted.

---

> > ### Author Response · Authors · 2025-10-15
> > **Response to Reviewer JhLE**
> >
> > Thank you for the thoughtful follow-up and for the positive recommendation. We have added the DMS-benchmarking limitation to the Conclusions and uploaded the revised manuscript. We appreciate your supportive review.

---

### Review · Reviewer_stWx · 2025-10-02

**Summary Of Contributions:**

Summary of Contributions

This paper presents FusionProt, a unified model that learns representations via iterative, bidirectional fusion between a protein language model and a structure encoder.

Strengths:
1. The paper introduces a novel fusion mechanism that leverages a learnable fusion token, which iteratively and bidirectionally integrates 1D sequence and 3D structural information.
2. Experiments are broad, covering EC, GO, and mutation stability tasks, with performance exceeding baselines. Ablation studies further analyze fusion frequency and the impact of different structure encoders.
3. The structure of the paper is clear, and the methodology is well-explained.

Weaknesses:
1. The design choices of the fusion token, particularly its universal connection to all nodes, are not deeply discussed and may raise concerns of redundancy.
2. The hierarchical interaction with the fusion token may incur additional computational and memory costs. Although the authors claim inference overhead is only 2–5%, detailed data is missing.
3. While improvements over baselines are reported, the performance gains appear modest in some cases, and statistical significance of these comparisons is not consistently provided.
4. Since the structural data may come from predicted structures (e.g., AlphaFold2), the model should be tested more rigorously under low-confidence or erroneous structural inputs to validate robustness.

**Additional Comments:**

No.

**Audience:**

Yes

**Audience Explanation:**

The integration of sequence and structural information is an active topic in protein machine learning, and the unified design with iterative cross-modal fusion is of clear interest to researchers in representation learning and bioinformatics. Even beyond protein-specific tasks, the token-based fusion strategy offers insights applicable to other multimodal learning settings.

**Broader Impact Concerns:**

No.

**Claims And Evidence:**

Yes

**Claims Explanation:**

Most major claims, such as performance improvements over baselines and the effectiveness of the proposed fusion mechanism, are supported by experimental evidence, including ablations and comparisons with alternative structure encoders. However, some claims extend beyond the evidence: assertions of capturing “biologically relevant features,” “strong generalization,” and “minimal overhead” are not fully quantified. Robustness to extreme cases (e.g., long proteins, low-quality structures, sparse labels) and scalability analysis are insufficiently demonstrated.

**Requested Changes:**

1. Provide a more detailed analysis of computational and resource costs, including inference time, memory use, and training efficiency, across different protein lengths and model scales.
2. Evaluate robustness under low-quality or uncertain structural inputs (e.g., low-confidence AlphaFold2 predictions), possibly through controlled ablation or threshold-based experiments.
3. Offer deeper comparative analysis with alternative design choices, such as static fusion models or unidirectional fusion, to clarify the specific advantages of the proposed mechanism.
4. Please consider extend evaluation to more challenging downstream tasks, such as extremely long proteins or low-quality structural datasets.
5. Add interpretability analyses, for example visualizing attention weights of the fusion token across layers, to provide more transparency.

---

> ### Author Response · Authors · 2025-10-09
> **Response to Reviewer stWx**
>
> We sincerely thank **Reviewer stWx** for the constructive review and for acknowledging the value of a unified, iterative fusion design. Your comments helped us sharpen the presentation, quantify costs more clearly, and surface robustness and interpretability analyses that improve transparency. We have addressed all requested changes and uploaded a revised manuscript.
>
> ---
>
> **General comment on language and scope**
>
> We revised the manuscript to avoid claims that are broader than the presented evidence. In particular, we now (i) qualify statements about “biologically relevant features”, framing them as hypotheses supported by observed task gains rather than causal conclusions (Supplementary Section A.2); and (ii) substantiate the “minimal overhead” claim by adding concrete throughput/memory measurements and complexity accounting (Supplementary Section A.1.5). We also ensured statistical reporting is consistent throughout (see Tables 1-5; where we validate statistical significat with a paired *t*-tests and Shapiro–Wilk test). We appreciate these suggestions; they helped us align claims tightly with evidence.
>
> ---
>
> **Requested Change 1**:
>
> We sincerely thank the reviewer for their thoughtful and constructive comment. We expanded **Supplementary Section A.1.5** with both theory and measurements to support the stated **2–5%** overhead and address inference time, memory, and training efficiency.
>
> ---
>
> **Requested Change 2**:
>
> We refer to **Section 5.4 “Robustness to Low-Confidence Regions in 3D Structures”**. To simulate uncertain inputs, we inject zero-mean Gaussian noise ($\sigma \in [0.1, 1.5]$ Å) into AF2 Cα backbones and report EC and GO-BP across noise levels. Performance is **stable up to $\sigma \approx 0.9$ Å**, with only a **3–5 point** absolute drop in $F_{\max}$ beyond this threshold. We clarified and expanded this section in the revision.
>
> ---
>
> **Requested Change 3**:
>
> We clarified in the **Introduction** and **Section 3.3.1 (“Fusion Design”)** that **ESM–GearNet** serves as our **static/one-shot fusion** baseline (independent encoders with late concatenation). In contrast, **FusionProt** employs a **dynamic fusion token** that is updated layer-by-layer via **bidirectional** interactions, enabling sequence and structure to co-adapt during training. We also expanded **Section 3.3.1** to motivate the token’s **universal connection** and added pointers to **Section 5.1 (“Ablation on Fusion-Injection Frequency”)**, which highlights the benefits of iterative (layerwise) fusion.
>
> ---
>
> **Requested Change 4**:
>
> We incorporated targeted tests in **Section 5.4** and **Section 5.5**:
> (i) **Low-confidence structures** via Gaussian backbone noise (0.1–1.5 Å) as above; stability holds up to ~0.9 Å with **3–5 points** drop thereafter on EC/GO-BP.
> (ii) **Length stratification & graph construction**: results are largely robust under our truncation protocol, with **longer proteins ($L \gtrsim 600$)** benefiting slightly from denser connectivity (**$k=12$** or **$d_{\mathrm{radius}}=12$ Å**; EC gains on the order of **+0.5 points, with p < 0.05**). We present these analyses on EC and GO-BP because, to our knowledge, there is not yet a commonly used public benchmark of extremely long or low-quality structures on which existing PLMs are routinely evaluated.
>
> ---
>
> **Requested Change 5**:
>
> This is an excellent suggestion. We added **Supplementary Section A.3** and a new **Figure 6** visualizing fusion-token attention across layers. The plots show fusion-token→residue attention (special tokens removed; row-normalized for visibility) and the token’s self-attention (original scale), with horizontal guides at each 5-layer structure injection. Across proteins, attention **sharpens with depth** and **refocuses after injections**, indicating that the token actively queries specific residues rather than primarily self-attending.
>
> ---
>
> **On statistical significance and “modest” gains**
>
> We ensured **significance markers are reported consistently in all tables**, and we verified differences using **standard paired *t*-tests**. For several tasks, even seemingly modest absolute gains correspond to meaningful effect sizes given the difficulty and maturity of these benchmarks.
>
> ---
>
> We appreciate the reviewer’s thorough analysis and actionable suggestions. In this revision, we added clearer cost accounting, robustness tests, comparative design context, and interpretability analysis, while addressing the reviewer's concerns. Thank you again for the thoughtful review and for helping us strengthen the work.

---

> > ### Comment · Reviewer_stWx · 2025-11-17
> >
> > Thanks for the responses! Most of my previous concerns have been addressed. I am happy to recommend the acceptance of the manuscript.

---

### Review · Reviewer_p7um · 2025-10-02

**Summary Of Contributions:**

The paper presents FusionProt, a simple yet general framework for integrating protein sequence and structure information through a single learnable “fusion token” that shuttles between a sequence language model (ESM-2-650M) and a structure encoder (default: GearNet). The authors evaluate FusionProt on enzyme commission (EC) prediction, Gene Ontology (GO: BP/MF/CC), and mutation stability prediction (MSP). Across these tasks, FusionProt outperforms strong sequence-only, structure-only, and prior joint models, with reported statistical significance.

## Strengths:
- **Clear Model Desgin**: A single fusion token provides a lightweight, model-agnostic mechanism to couple existing sequence and structure encoders.
- **Comprehensive Experiments**: Improvements are demonstrated across multiple benchmarks (EC, GO, MSP) with replicated runs and significance testing. Meanwhile, the ablation study is also extensive.

## Weakness:
- **Insufficient literature review: ** Several recent multimodal protein models are not cited / discussed.
- **Novelty issue of "fusion token": ** The idea of the "fusion token" is not new. Some related studies should be discussed.

**Additional Comments:**

N/A

**Audience:**

Yes

**Audience Explanation:**

Protein ML is a fast-moving, high-impact application area—energized by AlphaFold/AlphaFold 3 and the rapid rise of protein language models—so a method that improves sequence–structure fusion is likely to interest part of the TMLR audience.

**Claims And Evidence:**

Yes

**Claims Explanation:**

This paper provides multi-benchmark gains with repeated runs and significance tests, which aligns with TMLR’s primary criterion that claims be backed by accurate, convincing, and clear evidence.

**Requested Changes:**

Add discussion and, where doable, numbers vs.  MIF / MIF-ST (seq+struct pretraining/fusion), and SSEmb / Residue-Level Alignment (joint seq-struct embedding). If benchmarking isn’t feasible, include an explicit discussion in related works.

Regarding the novelty of the fusion token, add deeper discussion on recent graph virtual node methods and graph prompt learning approaches.

Ablate radius/k-NN/edge types and summarize trends vs. protein length/domain count (curves + recommended ranges).






- [MIF / MIF-ST] Masked inverse folding with sequence transfer for protein representation learning
- [SSEmb] SSEmb: A joint embedding of protein sequence and structure enables robust variant effect predictions

---

> ### Author Response · Authors · 2025-10-09
> **Response to Reviewer p7um**
>
> We sincerely thank **Reviewer p7um** for their detailed feedback and insightful suggestions, which have improved the clarity and presentation of our work. We have addressed all requested changes and uploaded a revised manuscript.
>
> ---
>
> **Requested Change 1**:
>
> Thank you for these suggestions. We have expanded **Related Work** (Section 2.3) to cover **MIF/MIF-ST** (sequence+structure pretraining/fusion) [1], **SSEmb** (a structure-graph GNN augmented with PLM features) [4], and **contrastive sequence–structure alignment** methods (**RLA** [3], **S-PLM** [2]). We clarify how these families differ from our design and where they tend to be strongest. To keep the benchmark focused, we report **MIF-ST** (which generally outperforms MIF) as the representative MIF variant in the main tables and retain **S-PLM** (which outperforms **RLA**) as the alignment baseline. We discuss **SSEmb** qualitatively and note that recent evaluations [5] place **SaProt-class** methods ahead on most tasks; thus, we prioritize SaProt-class and alignment baselines quantitatively.
>
> **Numbers we added.** We report quantitative comparisons:
>
> | Method          |   EC   | GO-BP | GO-MF | GO-CC |
> |---|---:|---:|---:|---:|
> | MIF-ST [1]      | 0.803 | 0.239 | 0.627 | 0.322 |
> | S-PLM [2]       | 0.885 | 0.470 | 0.674 | 0.460 |
> | **FusionProt**  | **0.904 ± 0.003** | **0.524 ± 0.004** | **0.689 ± 0.002** | **0.518 ± 0.004** |
>
> Differences versus baselines were confirmed using standard paired *t*-tests.
>
> **References:**
>
> [1] Masked inverse folding with sequence transfer for protein representation learning.
>
> [2] S-PLM: Structure-aware Protein Language Model via Contrastive Learning between Sequence and Structure.
>
> [3] Jointly Embedding Protein Structures and Sequences through Residue Level Alignment.
>
> [4] SSEmb: A joint embedding of protein sequence and structure enables robust variant effect predictions.
>
> [5] Exploring zero-shot structure-based protein fitness prediction.
>
> ---
>
> **Requested Change 2**:
>
> We added a focused paragraph in **Section 3.3.1** situating our **fusion token** alongside virtual nodes and graph prompts:
>
> - **Virtual nodes** aggregate and broadcast a **graph-global** summary *within a single modality* (the graph) and do **not** transport information between modalities.
>
> - **Graph prompts** prepend learned tokens to steer a *single-modality* GNN or pretrained graph model; prompts are typically static once injected.
>
> **Our design.** The FusionProt token **shuttles across modalities at every layer**, participating in (i) the PLM’s self-attention and (ii) the GNN’s message passing, carrying sequence cues into structure and structure cues back into sequence while preserving modality-specific inductive biases (sequence order; 3D geometry). Thus, the token is **instance-conditioned** and **layer-wise updated** by *both* modalities—distinct from graph-only virtual nodes and static prompts. To the best of our knowledge, prior PLMs do not implement **per-layer, bidirectional** sequence–structure fusion via a single token that participates in both attention and message passing.
>
> ---
>
> **Requested Change 3**:
>
> We added **Section 5.5** and **Figure 2** with stratified curves by sequence length, varying:
>
> - $k \in \lbrace 8,10,12 \rbrace$ (KNN)
>
> - $d_{\mathrm{radius}} \in \lbrace 8,10,12 \rbrace$ Å
>
> **Findings.**
>
> - Performance is **stable** for $k \in \{8,10,12\}$ with a broad optimum near our provided settings: **$k=10$** and **$d_{\mathrm{radius}}=10$ Å**.
>
> - **Longer proteins** ($L \ge 600$) show small, consistent gains with denser graphs (either **$k=12$** or **$d_{\mathrm{radius}}=12$ Å**).
>
> **Recommended ranges.**
>
> - **Short/Medium ($L<600$)**: KNN with **$k \in \{8,10\}$** *or* radius with **$d_{\mathrm{radius}} \in \{8,10\}$ Å**.
>
> - **Long ($L \ge 600$)**: consider **$k=12$** *or* **$d_{\mathrm{radius}}=12$ Å**.
>
> - **Default used** in the paper: **$k=10$**, **$d_{\mathrm{radius}}=10$ Å**.
>
> ---
>
> We hope these revisions and clarifications have fully addressed the reviewer’s concerns. We thank the reviewer again for their constructive feedback.

---

> > ### Comment · Reviewer_p7um · 2025-11-16
> > **Thanks.**
> >
> > After reading the authors’ response, I am satisfied with the changes they have made. Therefore, I recommend accepting this paper.

---

### Decision · Action_Editor_HRyH · 2025-11-23

**Recommendation:** Accept as is

**Additional Comments:**

The revisions have effectively addressed prior concerns, including an expanded literature review on multimodal protein models (e.g., comparisons to graph virtual nodes and prompt learning), deeper discussion of the fusion token's design (e.g., universal connections and potential redundancy), and additional robustness tests under low-confidence AlphaFold2 structures. The paper is now well-positioned for publication, with clear writing, intuitive methodology, and accessible background explanations. Minor suggestions for polish include clarifying the relation between consecutive subsequences and edge masking in the contrastive loss appendix. Overall, this is a solid contribution that advances efficient fusion strategies for biological multimodal data.

**Audience:**

Yes

**Audience Explanation:**

Protein representation learning is a core challenge in computational biology and machine learning, with broad implications for downstream tasks in drug discovery, enzyme engineering, and structural bioinformatics. FusionProt's lightweight, model-agnostic approach to multimodal integration appeals to ML practitioners working on foundation models, graph neural networks, and biological sequence/structure modeling, making it relevant to TMLR's interdisciplinary readership.

**Claims And Evidence:**

Yes

**Claims Explanation:**

The authors provide comprehensive experimental evidence across multiple benchmarks (EC prediction, GO annotation, and mutation stability prediction), including replicated runs, ablation studies on fusion frequency and structure encoders, and statistical significance testing where improvements are reported. The revisions have further strengthened this by adding detailed discussions of the multi-view contrastive loss and addressing computational overhead (e.g., confirming 2–5% inference increase with supporting data). While performance gains are modest in some cases, they are consistently demonstrated over strong baselines (sequence-only, structure-only, and prior joint models like ESM-GearNet), with clear motivation for the fusion token's bidirectional shuttling mechanism.

---

> ### Author Response · Authors · 2025-11-26
> **Official Comment by Authors**
>
> Dear Action Editor and Reviewers,
>
> We sincerely thank the Action Editor and Reviewers for their constructive feedback, which helped us improve our work. In the camera-ready version, we de-anonymized the paper and clarified the relationship between subsequence cropping and edge masking in the multiview contrastive objective, following the editor’s suggestion. Additionally, we have included all relevant code and datasets to ensure reproducibility.
>
> Sincerely,
>
> The Authors